# Foundation Models as Physical Priors: Decoupling Geometric Reasoning from Small-Molecule Solubility Prediction

**Jonathan Broadbent**
Sanofi, Digital R&D
Toronto, ON M5V 0E9

**Michael Bailey**
Sanofi, Digital R&D
Toronto, ON M5V 0E9

**Mingxuan Li**
Sanofi, Digital R&D
Cambridge, MA 02141

**Abhishek Paul**
Sanofi, CMC Synthetics
Cambridge, MA 02141

**Louis De Lescure**
Sanofi, CMC Synthetics
Cambridge, MA 02141

**Paul Chauvin**
Sanofi, Digital R&D
Barcelona, 08016, Spain

**Lorenzo Kogler-Anele**
Sanofi, Digital R&D
Toronto, ON M5V 0E9

**Yasser Jangjou**
Sanofi, CMC Synthetics
Cambridge, MA 02141

**Sven Jager**
Sanofi, Digital R&D
Frankfurt, 65929, Germany
sven.jager@sanofi.com

## Abstract

Scientific foundation models offer a promising path to generalized physical reasoning, yet integrating them into specific property prediction tasks remains an open architectural challenge. We evaluate two paradigms for leveraging geometric foundation models in solution-phase chemistry: (1) an end-to-end strategy, introducing Solvaformer, a hybrid SE(3)-equivariant transformer trained to learn geometric interactions from scratch, and (2) a decoupled strategy, where a pre-trained interatomic potential (AIMNet2) serves as a frozen feature engine for a lightweight scalar network. Evaluating on a massive combined dataset of quantum-mechanical and experimental solubility (BigSolDB 2.0), we find that the decoupled approach outperforms the bespoke end-to-end architecture while offering superior training efficiency. Our results suggest that scientific foundation models are most effective when used as composable physical priors—offloading complex geometric reasoning to specialized pre-trained backbones while allowing downstream models to focus on task-specific correlations. This modular "Simplicity at Scale" paradigm offers a robust blueprint for integrating classical scientific tools with modern deep learning.

## 1 Introduction

The rise of scientific foundation models presents a new architectural question for chemistry and materials science: how should we integrate general-purpose physical reasoning into specific property prediction tasks? While standard practice often favors training massive end-to-end models from scratch, this approach is computationally expensive and data-hungry. An alternative, modular paradigm is emerging: using pre-trained foundation models—such as Machine Learning Interatomic Potentials (MLIPs)—as "physical priors" that feed into lightweight downstream networks.

We investigate this trade-off in the context of small-molecule solution-phase chemistry, a domain critical for pharmaceutical manufacturing but historically challenging for predictive modeling due to data scarcity and complex geometric interactions (de Ruyter et al., 2022). Accurate small-molecule solubility prediction requires capturing subtle steric and electrostatic effects, traditionally the domain of expensive Density Functional Theory (DFT) simulations.

Existing approaches to solubility prediction span a spectrum from physics-based to purely data-driven. DFT-based models (e.g., COSMO-RS (Klamt, 2005)) provide highly accurate thermodynamic

predictions but require up to 10 hours per molecule for conformer generation and optimization, rendering them impractical for high-throughput screening (Kastenholz & Hünenberger, 2006). In contrast, Message Passing Neural Networks (MPNNs) (Gilmer et al., 2017) achieve competitive performance by operating on 2D molecular graphs, offering scalability but lacking explicit 3D reasoning and interpretability. To bridge these gaps, we build upon Equiformer (Thomas & Smidt, 2022), an SE(3)-equivariant graph transformer that respects rotational and translational symmetries through spherical harmonic representations (see Appendix D.3 for a mathematical description). This architecture provides a foundation for our Solvaformer design, which we adapt to handle multi-component solution-phase system.

In this work, we compare two strategies for scaling geometric reasoning. First, we introduce Solvaformer, an end-to-end hybrid graph transformer that learns SE(3)-equivariant geometric interactions directly from data. Second, we evaluate a decoupled strategy, where we utilize a pre-trained foundation model (AIMNet2) solely as a feature engine to generate physics-informed descriptors for a simple scalar network. We demonstrate that this decoupled approach not only matches the fidelity of computationally intensive DFT baselines but also outperforms the bespoke end-to-end architecture. Our findings support a modular "Simplicity at Scale" design principle for AI in science: leveraging foundation models as frozen reasoning engines to enable scalable, composable, and accurate downstream predictions.

## 2 METHODS

### 2.1 DATA

To balance experimental relevance with computational scale, we utilized a combined dataset of experimental solubility measurements and quantum-mechanical calculations. Our primary source is BigSolDB 2.0 (Krasnov et al., 2025), comprising 103,944 experimental LogS values across 213 solvents. To enable generalization across broader chemical space, we co-trained on CombiSolv-QM (Vermeire & Green, 2020), which provides 1 million theoretical solvation free energies ($\Delta G_{\text{solv}}$) computed via COSMO-RS. This multi-task regime allows models to learn from high-fidelity but sparse experimental data while leveraging the massive scale of physics-based simulations.

### 2.2 END-TO-END GEOMETRIC LEARNING (SOLVAFORMER)

Our first strategy investigates the "End-to-End" paradigm: can a single unified architecture learn both the rigid quantum-mechanical properties of solutes and the fluid thermodynamics of solvation? We introduce Solvaformer, a hybrid SE(3)-equivariant transformer that extends EquiformerV2 to multi-component systems (Figure 1). Unlike standard geometric models that enforce global equivariance, Solvaformer recognizes that while intramolecular geometry is rigid (requiring SE(3) constraints), the relative solute-solvent orientation is stochastic.

To capture this physical reality end-to-end, we enforce independent SE(3) symmetries for each molecule. We apply computationally intensive equivariant attention only to intramolecular atoms (to learn sterics and electronic structure) while using efficient scalar attention for intermolecular interactions (to learn solvation thermodynamics). This design attempts to bake physical inductive biases directly into the model weights.

Equivariant and scalar attention modules aggregate messages from within-molecule and from other-molecules. Suppose $i$ indexes a destination atom, $j$ indexes another atom in the same molecule, and $\zeta$ indexes an atom in the other molecule, with embeddings $x_i$, $x_j$, and $x_\zeta$. To compute the message incident on atom $i$, we compute the messages from equivariant attention and scalar cross-attention, and then sum them. For the equivariant attention, Equiformer computes message tensor values $v_{ij}^e$ using tensor products between $x_i$ and $x_j$, and similarly projects from $x_i \otimes x_j$ down to scalar features $f_{ij}^{(0)}$, which it then passes through a layer norm and activation to produce a logit

$$z_{ij}^e = \text{LeakyRELU}(\text{LayerNorm}(f_{ij}^{(0)})). \tag{1}$$

The scalar cross-attention is a simple implementation of traditional dot product attention, with the message values $v_\zeta^s$, and the key and query vectors, $k_\zeta$ and $q_i$ computed from the scalar part of the embedding, $x_\zeta^{(0)}$, by linear maps. Then the logits are $z_{i\zeta}^s = \langle q_i, k_\zeta \rangle$. The messages are aggregated

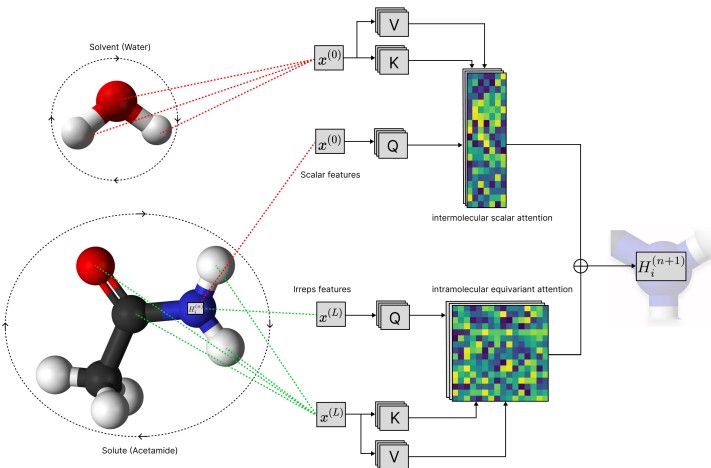

Figure 1: Example of Solvaformer performing an update of the hidden representation of a nitrogen atom ($H_i$) in a single layer for inputs water and acetamide.

the same way for both, with the caveat that equivariant messages are tensorial, while scalar messages are purely scalar quantities:

$$m_i^e = \sum_j \text{softmax}_j(z_{ij}^e) \ v_{ij}^e \qquad m_i^s = \sum_\zeta \text{softmax}_j(z_{i\zeta}^s) \ v_{i\zeta}^s \tag{2}$$

$$\mu_i = m_i^e + m_i^s \tag{3}$$

## 2.3 DECOUPLED REASONING WITH FOUNDATION MODELS

Our second strategy decouples geometric reasoning from the prediction task. We adopt a standard Message Passing Neural Network (MPNN) architecture (Gilmer et al., 2017) consisting of a message-passing phase, an interaction phase (resolving solute-solvent mappings via matrix multiplication), and a prediction phase. While the base MPNN operates on 2D graphs, we augmented it with "physical priors" derived from a pre-trained foundation model.

We utilized the NVIDIA ALCHEMI microservice to deploy the AIMNet2 backbone (Anstine et al., 2025)—a general-purpose machine learning interatomic potential (MLIP)—to predict partial atomic charges for all molecules. AIMNet2 serves as a frozen feature engine: it initializes node features with rich geometric information (interatomic distances and $l = 1$ harmonics) but compresses them into high-fidelity invariant scalars (charges) during inference. These descriptors are then concatenated with the MPNN's standard features. This approach effectively offloads the burden of learning electronic structure to the foundation model, allowing the lightweight MPNN to focus solely on learning solubility correlations without the computational overhead of explicit equivariant layers.

We also leveraged the NVIDIA ALCHEMI microservice to augment Solvaformer with solution-phase conformer ensembles. Rather than relying on single conformers generated via standard RDKit MMFF minimization, we utilized the batched geometry relaxation (BGR) microservice, which is powered by the pre-trained AIMNet2 backbone. Our protocol begins by generating an initial pool of 128 conformers using RDKit, which are subsequently filtered to ensure structural diversity. We then query the BGR microservice to relax a combined geometry containing the solute interacting with two to three explicit solvent molecules, effectively capturing the solute's conformational behavior within a simulated solution-phase environment. Following this step, the ensemble is filtered to retain only the ten lowest-energy conformations. During model training, these conformers are sampled according to their Boltzmann weights. Finally, the resulting forces and partial charges derived from the AIMNet2 model are directly incorporated as feature inputs into the initialization layer of Solvaformer.

Table 1: Model performance metrics on BigSolDB 2.0 test set

| Model | Geometric features | Paradigm | MAE | MSE | RMSE | $R^2$ | Pearson | Spearman |
|---|---|---|---|---|---|---|---|---|
| *XGBoost-DFT* | ✓ | Decoupled | 0.621 | 0.650 | 0.806 | 0.499 | 0.722 | 0.722 |
| SolvBERT | ✗ | End-to-End | 0.871 | 1.231 | 1.110 | 0.052 | 0.247 | 0.222 |
| XGBoost-CFP | ✗ | Decoupled | 0.749 | 0.889 | 0.943 | 0.315 | 0.592 | 0.561 |
| XGBoost-MMB | ✗ | Decoupled | 0.710 | 0.831 | 0.912 | 0.360 | 0.616 | 0.591 |
| MPNN | ✗ | End-to-End | 0.668 | 0.746 | 0.864 | 0.425 | 0.724 | 0.693 |
| Solvaformer | ✓ | End-to-End | 0.643 | 0.700 | 0.837 | 0.460 | 0.694 | 0.677 |
| MPNN w/ MLIPs | ✓ | Decoupled | 0.629 | 0.667 | 0.817 | 0.486 | 0.721 | 0.710 |
| Solvaformer w/ BGR | ✓ | Decoupled | 0.606 | 0.631 | 0.795 | 0.514 | 0.729 | 0.690 |

IMA: intermolecular attention; MLIP: machine learning interatomic potentials; BGR: batched geometry relaxation

## 3 RESULTS

Table 1 summarizes the performance of the evaluated paradigms. We include *XGBoost-DFT* as a gold-standard baseline; its decoupled approach (using expensive DFT descriptors) achieves the lowest error (MAE 0.621, RMSE 0.806) but is computationally prohibitive.

**End-to-End Learning (Implicit Physics).** Models attempting to learn chemical representations end-to-end exhibited varying performance. *SolvBERT* yielded the highest error (MAE 0.871), indicating that sequence-based foundation models may lack the physical grounding required for this thermodynamic task. The standard *MPNN* demonstrated competitive performance for a 2D scalar model (MAE 0.668), outperforming the sequence-based approaches. Introducing explicit geometry in the **Solvaformer** architecture led to further accuracy gains (MAE 0.643, RMSE 0.837), indicating that SE(3)-equivariant attention captures essential steric and electronic effects.

**Decoupled Reasoning (Explicit Physics).** Strategies that decoupled feature generation from prediction consistently outperformed their end-to-end counterparts. *XGBoost-MMB* (using MegaMolBART embeddings) achieved an MAE of 0.710, improving upon the end-to-end language model. By offloading geometric reasoning to the pre-trained AIMNet2 foundation model, *MPNN w/ MLIPs* achieved an MAE of 0.629, improving upon the standard end-to-end MPNN. The lowest error across all evaluated methods was achieved by **Solvaformer w/ BGR** (MAE 0.606, RMSE 0.795, $R^2$ 0.514). By coupling the Solvaformer architecture with explicit solution-phase geometric ensembles relaxed via the AIMNet2 foundation model, this decoupled approach surpasses the *XGBoost-DFT* baseline.

### 3.1 RUNTIME ANALYSIS

We evaluated the computational scalability of the different modeling paradigms by measuring the end-to-end inference latency (from SMILES to predicted LogS, including conformer generation) across varying molecular sizes and batch sizes. A detailed description of the experimental setup and profiling methodology is provided in Appendix B.

Our runtime analysis reveals critical scalability trade-offs between end-to-end learning, decoupled foundation models, and classical physics-based approaches. The gold-standard baseline, XGBoost-DFT, requires between $10^3$ and $10^4$ seconds per prediction (approximately 30 minutes to 3 hours). This latency is dominated by computationally demanding quantum chemistry preprocessing and exhibits exponential scaling with molecular size, rendering the DFT approach intractable for high-throughput synthesis optimization.

In contrast, the deep learning paradigms offer massive accelerations, though their efficiency diverges based on their reliance on explicit geometric features. The pure end-to-end models (Solvaformer and MPNN) maintain the lowest inference latencies among the neural architectures. By learning

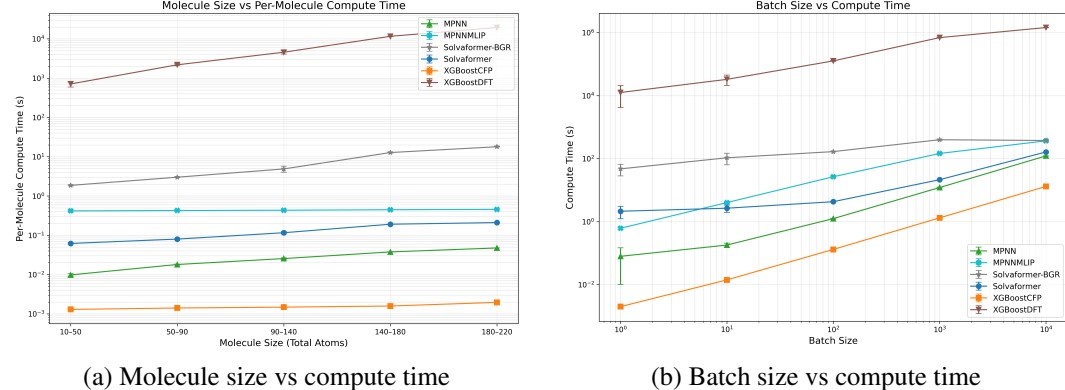

(a) Molecule size vs compute time      (b) Batch size vs compute time

Figure 2: Runtime analysis of compute time for small-molecule solubility prediction methods. Each method was computed on 4 CPUs and 1 Nvidia H100 GPU (80GB)

implicit representations directly from data without the bottleneck of external physical simulations, their compute times increase only marginally as molecular size scales.

The decoupled foundation model paradigms (Solvaformer-BGR and MPNN with MLIPs) introduce an intermediate computational cost. Because these models offload geometric reasoning to a pre-trained physical prior (AIMNet2) to generate explicit solution-phase features or partial charges, they incur a measurable overhead during inference, averaging between $10^0$ and $10^1$ seconds per molecule. However, both the decoupled and end-to-end neural models are highly GPU-accelerated and leverage batching effectively—runtimes remain flat from batch sizes 1 to 10 due to near-optimal GPU occupancy, scaling linearly only beyond the models' native capacities. Ultimately, while the decoupled methods introduce slight latency compared to their pure end-to-end counterparts, they successfully capture high-fidelity geometric priors while remaining orders of magnitude faster than intractable DFT simulations.

## 3.2 EXPLAINABILITY CASE STUDY: DISTINGUISHING INTRA- VS. INTERMOLECULAR HYDROGEN BONDS

A key advantage of the end-to-end Solvaformer architecture over decoupled scalar models is interpretability. We analyzed the model's intermolecular attention maps for two isomers in water: salicylic acid (ortho) and 4-hydroxybenzoic acid (para). This pair presents a classic geometric challenge: the para isomer's hydroxyl proton (H15) is exposed for hydrogen bonding, while the ortho isomer's H15 forms an internal bond with its own carbonyl group, reducing solvent interaction.

Solvaformer correctly recovers this mechanism (Figure 3). For the para isomer, the attention map shows strong interaction between H15 and water, identifying it as an active bonding site. In contrast, the map for salicylic acid (ortho) shows effectively zero attention from H15 to the solvent, correctly implying that the proton is "occupied" by the intramolecular bond. This demonstrates that while the decoupled strategy (MPNN w/ MLIPs) is efficient, the end-to-end geometric approach uniquely enables mechanistic auditing of chemical reasoning.

## 4 DISCUSSION

Our results offer a clear perspective on integrating foundation models into scientific workflows. The superior performance of decoupled strategies—most notably *Solvaformer w/ BGR*—demonstrates that purely end-to-end learning is not the optimal path for small-molecule solubility prediction. Instead of solely training bespoke architectures to relearn fundamental physics from scratch, it is highly effective to augment them with a pre-trained geometric foundation model (AIMNet2) acting as a reasoning engine. This composable approach allows us to leverage the massive pre-training of the foundation model to generate explicit solution-phase geometric priors. Consequently, the downstream predictor achieves exceptional accuracy, even surpassing computationally demanding DFT calculations, while remaining lightweight and scalable.

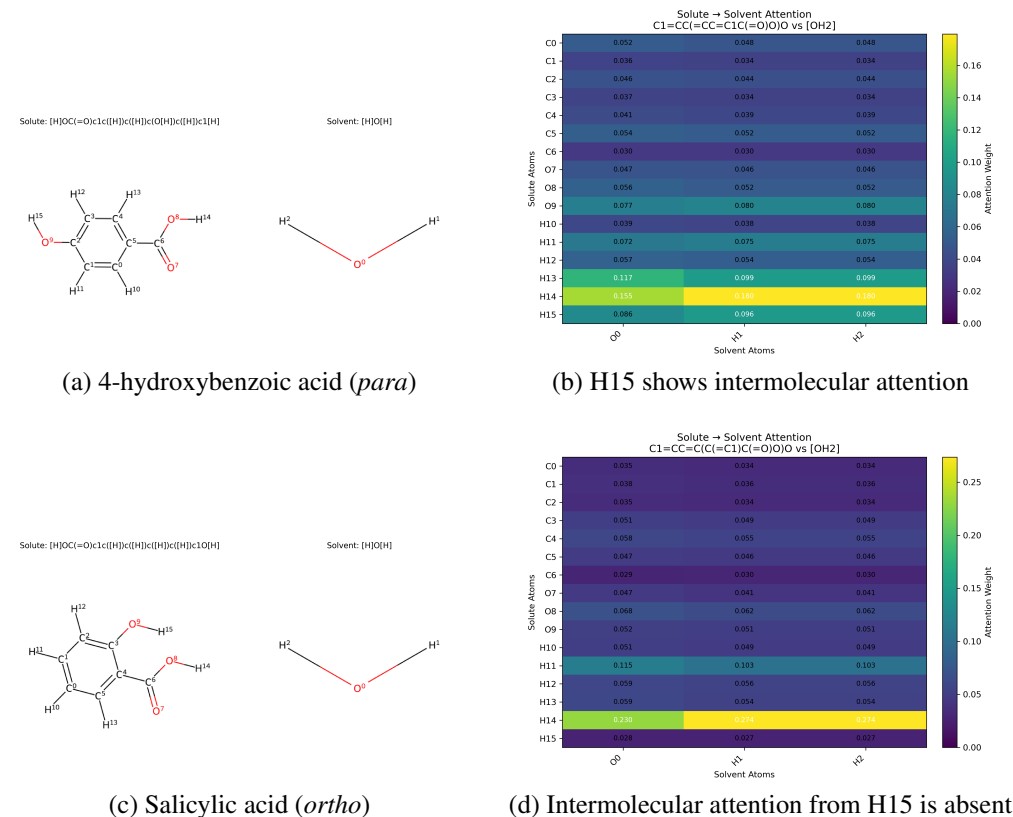

(a) 4-hydroxybenzoic acid (*para*)

(b) H15 shows intermolecular attention

(c) Salicylic acid (*ortho*)

(d) Intermolecular attention from H15 is absent

Figure 3: Solute-to-solvent attention maps demonstrating Solvaformer's chemical intuition. (b) For the *para* isomer, the hydroxyl proton (H15) shows clear attention to water, indicating intermolecular H-bonding. (d) For the *ortho* isomer, this attention from H15 disappears, correctly implying it is occupied in a dominant intramolecular H-bond.

Previously, this integration presented a distinct trade-off: decoupled scalar pipelines (such as *MPNN w/ MLIPs*) were computationally efficient and accurate, but lacked the mechanistic interpretability of pure end-to-end geometric models. Because the base Solvaformer models interactions explicitly through attention, it allows researchers to visualize chemically significant events—such as hydrogen bonding networks (Section 3.2)—that black-box scalar descriptors cannot reveal. By employing *Solvaformer w/ BGR*, we effectively resolve this dichotomy. This paradigm offloads the complex task of solution-phase geometry relaxation to the decoupled foundation model while retaining the geometric transformer for the final prediction. Ultimately, this demonstrates that combining frozen physical priors with interpretable attention architectures provides both state-of-the-art predictive fidelity and essential mechanistic insight.

Despite the clear predictive and mechanistic advantages of this hybrid approach, the pragmatic benefits of purely end-to-end architectures remain highly relevant. While *Solvaformer w/ BGR* establishes a new benchmark for accuracy, it inherently introduces pipeline complexity. Utilizing a secondary foundation model necessitates additional infrastructure, external microservice dependencies, and the computational overhead of dynamic geometry relaxation. In contrast, a purely end-to-end model like the base *Solvaformer* consolidates the entire workflow into a single, self-contained forward pass. This unified architecture requires significantly fewer computational resources to employ, involves less engineering setup, and maintains a strictly lower inference runtime. Therefore, while decoupled reasoning provides state-of-the-art fidelity, pure end-to-end learning remains an indispensable paradigm for environments where operational simplicity and near-instantaneous predictions are paramount.

**Limitations.** Our evaluation is constrained by the quality of the underlying data. The measured solubility labels are drawn from a meta-analysis of external measurements, introducing heterogeneity

from differing experimental protocols. As shown in Figure 4, individual measurements carry inherent experimental error which propagates into both training targets and evaluation metrics, potentially creating a performance ceiling that masks smaller model improvements.

**Takeaway.**    This study addresses the workshop's theme of "Rethinking Foundation Models for Science" by demonstrating that composability is a key design principle for modeling solution-phase chemistry. We show that the complex geometric reasoning required to capture solvation effects (e.g., steric hindrance, polarization) need not be learned entirely from scratch within every task-specific architecture. Instead, by offloading dynamic geometry relaxation to specialized pre-trained potentials (MLIPs) and integrating these physical priors into a geometric transformer (Solvaformer w/ BGR), we can surpass the accuracy of computationally prohibitive DFT calculations. Crucially, this composable approach resolves the historical dichotomy between accuracy and explainability; it achieves state-of-the-art predictive fidelity while retaining the attention mechanisms necessary to explicitly visualize specific solute-solvent interactions, such as hydrogen bonding networks. However, this integrated performance introduces pipeline complexity. Thus, for small-molecule solubility prediction, the optimal choice depends on the operational context: deploy composable foundation pipelines when maximizing accuracy and mechanistic insight is paramount, but utilize self-contained, end-to-end architectures when operational simplicity, minimal setup, and near-instantaneous inference are the primary constraints.

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

## A  SOLVAFORMER ABLATION STUDY

Solvaformer extends the Equiformer V2 architecture to multi-component systems, requiring a mechanism to model interactions between distinct solute and solvent molecules. Because solution-phase molecules are dynamic, their relative distances and orientations are stochastic rather than fixed.

To determine the optimal Intermolecular Attention (IMA) strategy, we evaluated three integrated configurations on the BigSolDB 2.0 test set (Table 2). The most effective approach was **Scalar IMA** (MAE 0.643, RMSE 0.837, $R^2$ 0.460), which utilizes standard scalar cross-attention. By dropping rigid equivariant constraints for intermolecular edges, the network implicitly learns thermodynamic interactions without requiring artificially specified orientations. Removing the attention mechanism entirely (**No IMA**), where solute and solvent node embeddings are pooled and aggregated independently, noticeably degraded predictive performance (MAE 0.691, $R^2$ 0.345). Finally, applying strict **Equivariant IMA**—by artificially sampling intermolecular distances and orientations from a normal distribution to initialize the attention scheme—yielded the poorest overall results (MAE 0.741, RMSE 0.949, $R^2$ 0.306).

These findings demonstrate a critical architectural distinction: while strict SE(3)-equivariant attention excels at capturing the rigid internal geometry of individual molecules, enforcing artificial spatial orientations between separate, dynamic molecules introduces detrimental noise. Consequently, a hybrid approach—utilizing equivariant attention for intramolecular mechanics and scalar attention for intermolecular thermodynamics—provides the optimal architecture for solubility prediction.

## B  MEASURING RUNTIME

To evaluate the scalability of the proposed paradigms, we conducted a comprehensive runtime analysis measuring end-to-end inference time—from SMILES string input to LogS output, inclusive of all

Table 2: Solvaformer ablation on BigSolDB 2.0 test set

| Intermolecular Attention | MAE | MSE | RMSE | $R^2$ | Pearson | Spearman |
|---|---|---|---|---|---|---|
| Equivariant | 0.741 | 0.900 | 0.949 | 0.306 | 0.672 | 0.640 |
| None | 0.691 | 0.858 | 0.926 | 0.345 | 0.626 | 0.606 |
| Scalar | 0.643 | 0.700 | 0.837 | 0.460 | 0.694 | 0.677 |

conformer generation and feature extraction steps. We evaluated representative models across our paradigms: end-to-end models (MPNN, Solvaformer), decoupled foundation models (MPNN with MLIPs, Solvaformer-BGR), and classical baselines (XGBoost-CFP, XGBoost-DFT).

Measurements were conducted using the BigSolDB 2.0 dataset, evaluating inference time as a function of both batch size and molecular size. For each data point, we sampled five independent repeats from the dataset, utilizing replacement when size constraints (e.g., molecules containing 180–220 atoms) yielded insufficient unique samples. We focused specifically on inference latency to simulate real-world discovery workflows where researchers require rapid predictions for high-throughput pathway optimization; training costs are considered a one-off computational investment and were excluded from this analysis.

To maximize GPU utilization without exceeding memory constraints on larger molecules, we employed a batch size of 32 for Solvaformer and 3000 for MPNN. For the Solvaformer-BGR pipeline, the recorded runtime incorporates the overhead of the NVIDIA ALCHEMI microservice workflow: generating an initial 128 RDKit conformers, filtering for structural diversity, relaxing the geometry via the AIMNet2 backbone, filtering for the lowest 10 energy conformations, and finally extracting the relevant forces and charges for the initialization layer. Lastly, to avoid excessive and unnecessary computational expenditure for the XGBoost-DFT baseline, we profiled five representative molecules spanning a range of molecular weights and fit an empirical scaling formula to estimate its preprocessing time based on the total solute-solvent molecular weight.

## C  DATA PREPROCESSING

### C.1  BIGSOLDB 2.0

We utilized the BigSolDB 2.0 dataset, a comprehensive solubility resource comprising 103,944 experimentally measured solubility values for 1,448 unique organic solutes in 213 solvents, across a temperature range of 243–425 K. These values were manually curated from 1,595 peer-reviewed publications and standardized into a machine-readable format including SMILES representations for both solutes and solvents. LogS values (log molar solubility in mol/L) were calculated using solvent densities either from experimental measurements or interpolated via linear models where necessary. The dataset spans aqueous and non-aqueous solvents, including common organic media such as ethanol, acetone, and ethyl acetate, enabling broad coverage for solubility prediction tasks (Krasnov et al., 2023; 2025).

To ensure the quality of the data used for model development, we applied the following filtering criteria:

- Canonicalized the SMILES of both solutes and solvents using RDKit.
- Removed entries containing bimolecular solutes or multi-component species.
- Excluded all metal-containing and ionic solute entries.
- Discarded entries lacking a LogS value.
- Discarded all duplicate entries.

The dataset had 6591 duplicated entries. LogS between duplicated entries had an average standard deviation of $0.0974$, which represents an intrinsic limit to the performance of our models (Figure 4). Following data filtering, we split the dataset into training and test sets using chemical space-aware clustering:

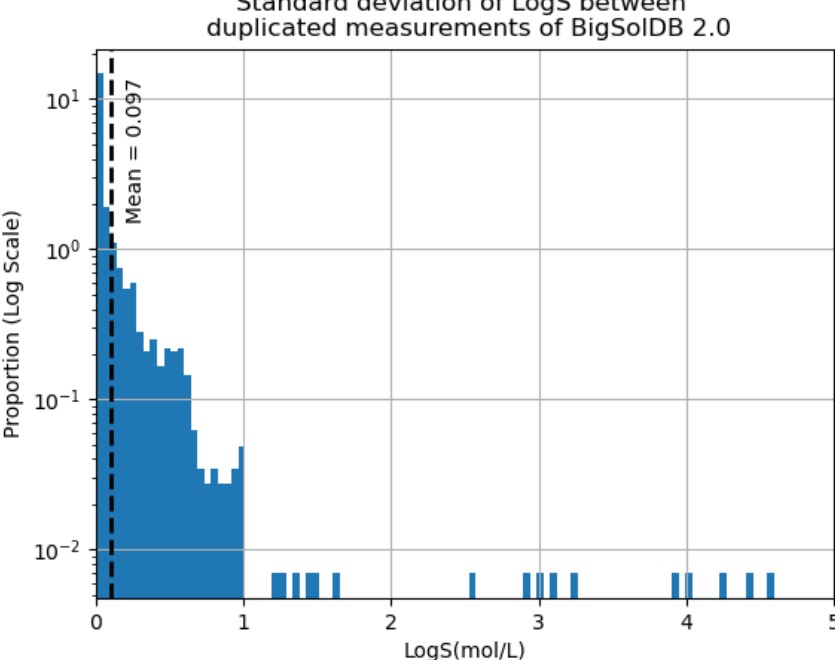

Figure 4: 6591 measurements in BigSolDB 2.0 had duplicated measurements from separate sources (same solute, solvent, temperature but measured in a different laboratory and have different measured solubility). We removed these measurements from the dataset. Here we measure the standard deviation within groups of duplicated measurements and plot the distribution. This provides an estimate of the precision of experimental measurements for solubility and hence lower bound for error rate prediction of our dataset.

- Solutes were clustered using the Butina algorithm based on Tanimoto similarity of Morgan fingerprints (radius = 2).

- From these clusters, we sampled solutes across the chemical space to form a structurally diverse test set (10% of data).

The final split consisted of 82,758 solute-solvent measurements for training and 9,250 for testing, with 1,142 unique solutes in the training set and 126 in the test set (See figures 6, 5).

This stratified, diversity-aware split enables robust benchmarking of model generalization to solute structures.

## C.2 COMBISOLV

The CombiSolv-QM dataset (Vermeire & Green, 2020) provides quantum-mechanically computed solvation free energies for approximately one million solvent–solute pairs. These values were derived using COSMO-RS theory via the COSMOtherm software, covering 11,029 solutes and 284 solvents. All solvation energies ($\Delta G_{\text{solv}}$) were calculated at 298 K using a conformer-aware protocol that includes DFT-based geometry optimization followed by chemical potential analysis in solution.

In our work, CombiSolv-QM was used in conjunction with the BigSolDB 2.0 dataset to train the Solvaformer model. To enable learning from both experimental and computational data, we implemented an alternating batch training scheme: each mini-batch was sampled from either CombiSolv-QM or BigSolDB 2.0. The model was trained with two separate prediction tasks: one for experimental solubility values (LogS) and one for calculated solvation free energies ($\Delta G_{\text{solv}}$). The training set of CombiSolv-QM was filtered for solutes in the BigSolDB 2.0 test set.

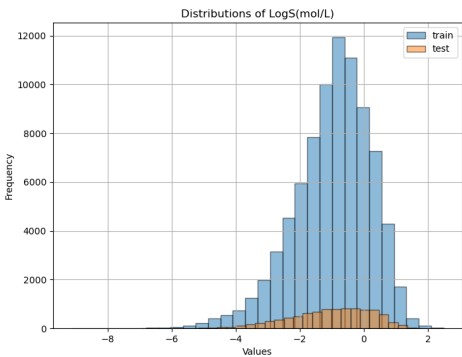

Figure 5: Distribution of measured logS in the train-test split.

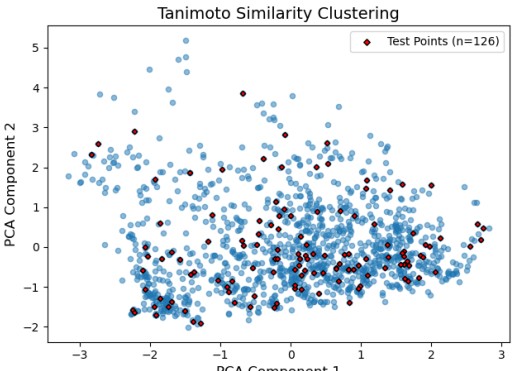

Figure 6: Butina clustering of tanimoto similarity of all unique solutes in BigSolDB 2.0

This dual-target, alternating-batch strategy allowed the model to generalize across both experimental and theoretical chemical space while maintaining fidelity to each target type. It also served as an effective form of multi-task learning, allowing the model to benefit from the scale and consistency of CombiSolv-QM and the real-world relevance of BigSolDB 2.0.

# D    ADDITIONAL MODEL DETAILS

## D.1    SOLVBERT

SolvBERT is a transformer-based model that treats solute–solvent complexes as combined SMILES sequences, applying NLP-style encoding to molecular interactions (Yu et al., 2023). Unlike graph-based models that process solute and solvent separately, SolvBERT ingests the concatenated SMILES of the complex and converts them into contextualized embeddings using a BERT backbone pre-trained in an unsupervised masked language modeling manner on a large computational dataset (CombiSolv-QM) (Yu et al., 2023). With this setup, the self-attention network is able to learn interactions between solute and solvent. Following pretraining, the model is fine-tuned either on experimental solvation free energy or solubility datasets, demonstrating strong performance across both tasks.

Empirical evaluations show that SolvBERT achieves solvation free energy predictive accuracy comparable to state-of-the-art graph-based models like MPNN. It also surpasses hybrid graph-transformer architectures such as GROVER when predicting solubility on out-of-sample solvent–solute combinations (Yu et al., 2023). The unsupervised pretraining enables better internal clustering of molecular systems (via TMAP visualization), supporting enhanced generalization despite varied fine-tuning targets.

## D.2    XGBOOST MODELS

We use XGBoost to predict solubility using a variety of embedding methods. For **DFT** features, we first generated conformers using RDKit ETKDGv3 through WEASEL 1.12. Conformers were then optimized using GFN2-xTB, and the five most stable conformers (or those covering 90% of the Boltzmann population at the xTB level) were subsequently subjected to DFT calculations at the wB97X-V/def2-TZVP level of theory in the gas phase. The energies derived from these DFT calculations were used to apply Boltzmann weights to the resulting molecular features. Features were calculated for both the solute and the solvent molecules. The most stable conformer was further evaluated with a COSMO-RS calculation using ORCA 6.0(Neese, 2012) to obtain the free energy of solvation.

We also generated **Circular Fingerprints (CFP; Morgan fingerprints)**(Morgan, 1965) with RDKit, using a radius of 2 and a fingerprint size of 2048 bits, with count simulation disabled, chirality excluded, bond types enabled, ring-membership information included, default count bounds, and without restricting to nonzero invariants.

In addition, we produced learned embeddings using **MegaMolBART (MMB)**, a large language model trained on 1.5 billion SMILES from the ZINC-15 dataset (Irwin et al., 2022; NVIDIA Corporation, 2024) and previously shown to be effective for molecular property prediction (Moayedpour et al., 2024). From the pretrained model we extracted 512-dimensional embeddings and used them directly as inputs to XGBoost.

Furthermore, the **SolvBERT** model was initially trained for temperature independent solubility. To adapt it for our purposes we first trained the model on the CombiSolv and BigSolDB 2.0 training data (Training followed the procedure described here: https://github.com/su-group/SolvBERT (Yu et al., 2023) and then took the embeddings from the pre-trained model and trained an XGBoost regressor with temperature as an additional feature to the model.

Using these feature sets, we trained a standard XGBoost regressor (Chen & Guestrin, 2016) with `n_estimators`=500, `learning_rate`=0.1, and `max_depth`=6.

Table 3: DFT calculated molecular features

| Feature name | Feature Description |
|---|---|
| Dipole_Moment_Debye | Dipole moment in Debye |
| LUMO_E_Eh | Energy of Lowest unoccupied molecular orbital (LUMO) in Hartrees |
| LUMOX_E_Eh | Energy of LUMO - X in Hartrees |
| HOMO_E_Eh | Energy of Lowest occupied molecular orbital (HOMO) in Hartrees |
| HOMOX_E_Eh | Energy of HOMO - X in Hartrees |
| HOMO_LUMO_gap | Energy difference between HOMO and LUMO |
| Dispersion_correction | Dispersion correction calculated with VV10 nonlocal van der Waals correlation |
| Cavity_Volume | CPCM cavity volume in cubic angstroms |
| Cavity_Surface_area | CPCM solvent-accessible surface in squared angstroms |
| Surface_Charge_CPCM | Total apparent surface charge distribution calculated by CPCM |
| C_charge_total | Sum of all Hirshfeld charge on all carbon atoms |
| O_charge_total | Sum of all Hirshfeld charge on all oxygen atoms |
| N_charge_total | Sum of all Hirshfeld charge on all nitrogen atoms |
| H_charge_total | Sum of all Hirshfeld charge on all hydrogen atoms |
| Het_charge_total | Sum of all Hirshfeld charge on all heteroatoms |
| energy_kcal_mol | Electronic energy of the system in kcal/mol |
| dGs | Free energy of solvation as calculated by open COSMO-RS through ORCA6 |

### D.3   EQUIFORMER

Equiformer is analogous to an ordinary graph transformer in the following sense:

- Instead of weights and activations taking scalar values, they take values in an SO(3) representation space. These representations are equivalent to *spherical harmonics* (also known as *orbitals*), so a weight or activation can be seen as an approximated function on the sphere $S^2$.
  - When a representation vector $f$ is decomposed into irreducible representations (i.e., different angular frequencies) $f_\ell$, its 'rotations' correspond to Wigner D-matrices:

  $$f_\ell \mapsto D^\ell(R) \, f_\ell$$

  - Multiplication of these SO(3) representations corresponds to multiplication of their spherical functions (dropping high-frequency terms where needed)
- In addition to taking non-scalar values, the weights are also spatially varying *functions*, depending on the relative vector between the communicating nodes. The spatial variation of these weights is also represented using a spherical harmonic decomposition, with radial dependence.
  - Therefore, the weight functions (and thus the model) are *equivariant* if rotating the evaluation vector in 3D space corresponds to "rotating" the weight value.

To compute the product of spherical functions $f$ and $g$ with harmonic decompositions $f_{\ell_1, m_1}$ and $g_{\ell_2, m_2}$, Equiformer uses tensor products based on Clebsch–Gordan coefficients (Sharp, 1960) $C^{\ell_3}_{\ell_1, \ell_2}$, which combine the components in the correct way:

$$[f_{\ell_1} \otimes g_{\ell_2}]_{\ell_3, m_3} = \sum_{m_1, m_2} C^{\ell_3, m_3}_{\ell_1, m_1; \, \ell_2, m_2} \, f_{\ell_1, m_1} \, g_{\ell_2, m_2}. \tag{4}$$

Of course, what distinguishes Equiformer from an ordinary equivariant message passing network is that Equiformer uses the above operations to build an equivariant attention mechanism, so that each node and head can pay different amounts of attention to different neighbor nodes.

EquiformerV2 (Liao et al., 2024) enhances the original Equiformer architecture. It replaces the SO(3)-equivariant convolutions with eSCN convolutions, reducing computational complexity from $O(L_{\max}^6)$ to $O(L_{\max}^3)$, enabling scaling to higher-degree ($L = 6$) representations (Passaro & Zitnick, 2023; Liao et al., 2024).

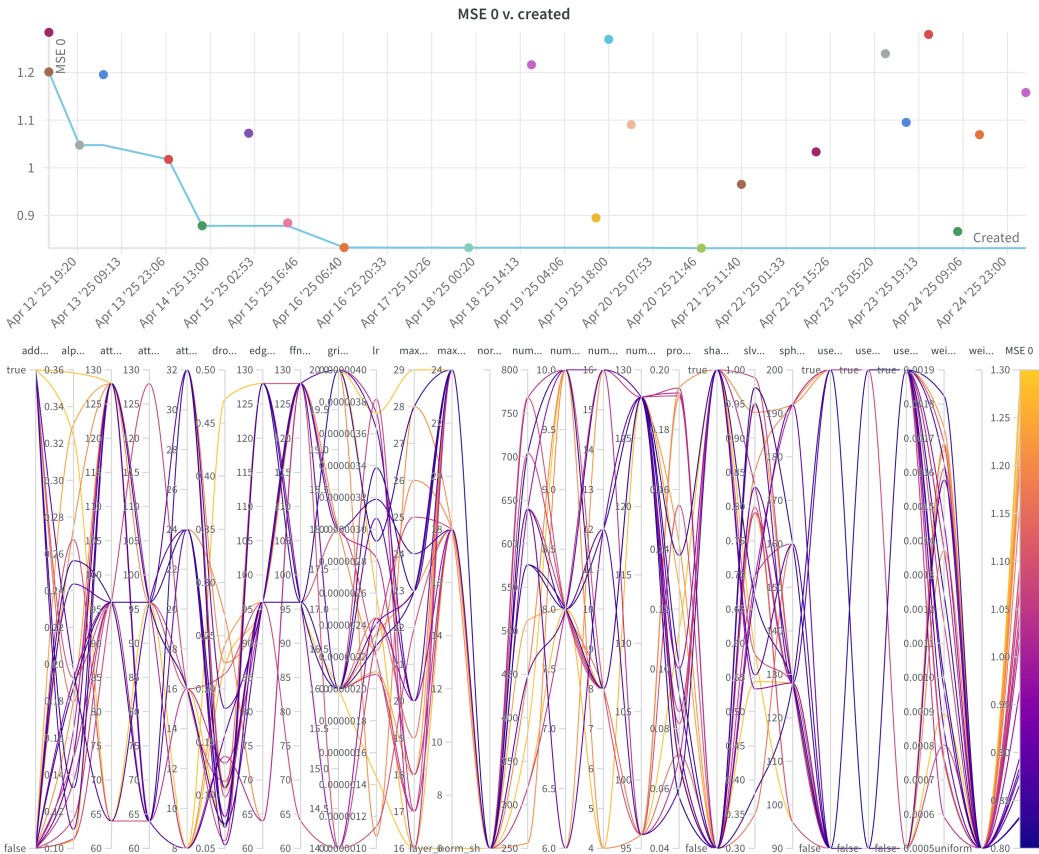

Figure 7: Hyperparameter tuning of Solvaformer. We ran a total of 23 different runs. WandB agents selected hyperparameters of successive runs using Bayesian optimization where performance was measured by MSE on the BigSolDB2.0 validation set.

EquiformerV2 achieved state-of-the-art results on large-scale datasets (e.g., OC20/OC22), which use force and energy of individual molecules as the training target. However, EquiformerV2 is not equipped to predict solubility.

## D.4 SOLVAFORMER TRAINING

We trained Solvaformer, using a combined dataset of BigSolDB 2.0 and CombiSolv-QM, sampled in equal ratios via alternating batches. 3D conformers were generated using RDKit and minimized by Merck molecular force field (MMFF). The model was trained with a batch size of 6 for up to 100 epochs, with early stopping based on a patience of 20 and a minimum delta of 0.01. Solvaformer consists of 8 layers with 8 attention heads, 128-dimensional spherical channels, and 96-dimensional hidden dimensions in both attention and feedforward networks. It uses SE(3)-equivariant operations with angular momentum up to $l = 6$, and includes solvent-solvent attention and edge features. Regularization includes alpha dropout (0.5), drop path (0.4), and projection dropout (0.4). The model predicts both solubility and solvation energy using separate outputs and is optimized with mean squared error loss and a learning rate of $3 \times 10^{-6}$. All hyperparameters were selected based on a hyperparameter optimization experiment, the details of which are provided in the appendix (Figure 7).

## E DATA AVAILABILITY

All the raw data used to train and test the models is publicly available and can be found here:

- BigSolDB2.0 (Krasnov et al., 2025)
     link: https://zenodo.org/records/15094979
     version: Published March 27, 2025 | Version v1
- CombiSolv-QM (Vermeire & Green, 2020):
     link: https://zenodo.org/records/5970538
     version: Published July 1, 2022 | Version v1.2

