# OpenReview forum: "Foundation Models as Physical Priors: Decoupling Geometric Reasoning from Small-Molecule Solubility Prediction"
_ICLR.cc/2026/Workshop/FM4Science — ICLR 2026 Workshop FM4Science Poster_

### Official Review · Reviewer_np4j · 2026-02-20
**Pretrained vs bespoke predictors in chemical deep learning**

**Rating:** 6
**Confidence:** 3

**Review:**

**Summary:**

The authors study whether simple prediction methods based on pretrained chemical foundation models can match or exceed the performance of task-specific (bespoke) neural architectures in chemical representation learning. Empirical findings suggest that pretrained foundation model-based predictors achieve higher accuracy on downstream tasks than specialized baselines.

**Points of Strength:**
1. The authors study an important question: whether/how much to rely on pretrained scientific foundation models in the chemical sciences for fundamental downstream tasks.

**Points for Improvement:**
1. Unclear computational complexity of Solvaformer: The authors do not discuss how their proposed changes (on top of Equiformer) change the time and memory complexity of the model. Discussing such details is important to help readers understand the applicability of Solvaformer in (large data) settings.
2. Unclear hyperparameter optimization of MPNN w/ MLIPs: The authors should discuss whether/how they tuned the hyperparameters of their simpler (foundation model-based) baseline.
3. Missing punctuation: E.g., period at the end of Line 105.

---

### Official Review · Reviewer_yFrc · 2026-02-21
**Review on Foundation Models as Physical Priors: Decoupling Geometric Reasoning from Small-Molecule Solubility Prediction**

**Rating:** 5
**Confidence:** 4

**Review:**

# Quality

## Pros:
The experimental comparison is well-structured and the data preprocessing is thorough with a Butina clustering split.
## Cons:
- Table 1 reports single-point metrics without error bars or confidence intervals.

# Clarity

The paper is generally well-written with clear motivation and logical structure. The mathematical formulation is concise for me. Figure 1 effectively illustrates the Solvaformer architecture, and the attention map case study (Figure 2) is a compelling visual.

## Some clarity issues:
- Appendix C lists the same Zenodo URL (`https://zenodo.org/records/15094979`) for both BigSolDB 2.0 and CombiSolv-QM, which appears to be an error.
- The term "Tiny Paper Track" appears in the header but this is submitted to the FM4Science Workshop — a formatting oversight.

# Originality

Extending EquiformerV2 to multi-component systems with separate equivariant intramolecular and scalar intermolecular attention is a meaningful and novel design.

# Significance

The finding from this paper that a simple decoupled approach outperforms a sophisticated end-to-end architecture is a valuable empirical data point.

# Pros
- Table 1 covers 8 models spanning both paradigms, including a gold-standard DFT baseline, language model (SolvBERT), traditional ML (XGBoost variants), and GNNs (MPNN, Solvaformer). This provides a thorough landscape of approaches. [VERIFIED]
- The ablation study (Section 3, Table 1) demonstrates that intermolecular attention is essential.
- Figure 2 (Section 3.1) provides a chemically convincing demonstration that Solvaformer learns meaningful hydrogen bonding patterns.
- Training on ~83K experimental measurements from BigSolDB 2.0 and 1M theoretical values from CombiSolv-QM provides a substantial evaluation setting.

# Cons
- The paper claims superior training efficiency (Abstract) for the decoupled approach but provides no quantitative comparison of training times, FLOPs, or GPU hours.
- Section 2.3 states the foundation model is used to predict partial atomic charges for all molecules, but AIMNet2 can produce energies, forces, Hessians, and other properties. The paper does not ablate which AIMNet2 features matter most or justify why only charges were selected.
- Insufficient engagement with directly related prior work.
- The attention map analysis (Section 3.1, Figure 2) covers only one pair of isomers. While chemically compelling, a single example is insufficient to support the claim of mechanistic auditing of chemical reasoning (Section 3.1).

---

### Official Review · Reviewer_fvVQ · 2026-02-22
**The conclusions are limited by the assumption of a single static 3D conformer and possible domain mismatch between the foundation model and new chemistry.**

**Rating:** 8
**Confidence:** 3

**Review:**

This paper studies how pre-trained scientific models should be used in downstream chemistry tasks, using small-molecule solubility prediction as an example. The authors compare two approaches: an end-to-end SE(3)-equivariant transformer (Solvaformer), which tries to learn geometric and electronic interactions directly from 3D molecular structures, and a decoupled strategy in which a pre-trained machine learning interatomic potential (AIMNet2) is used as a frozen “physical prior,” and its outputs are fed into a simpler prediction model. The authors show that the decoupled approach is more computationally efficient, while the end-to-end Solvaformer provides greater mechanistic interpretability. The study is carefully conducted, includes strong baseline comparisons and tests showing which parts of the model contribute most to performance, and addresses a timely architectural question: whether physics-intensive tasks benefit more from reusing pre-trained scientific models in a modular way than from training large task-specific models from scratch.
However, several assumptions limit the scope of the conclusions. First, the method relies on a single static 3D conformer, which is a strong simplification for solubility and may not generalize to highly flexible molecules, where conformational ensembles and entropic effects are important. Second, there is a potential domain gap concern: if the foundation model (e.g., AIMNet2) was not trained on the specific types of chemistry or molecular regimes relevant to a new application, its “physical prior” may not transfer reliably. Therefore, the broader claim that scientific foundation models are best used as composable physical priors may depend strongly on how well the training domain matches the target chemistry, as well as on the rigidity of the molecules being studied. Overall, the paper provides a clear and practically useful architectural insight, but its impact would be strengthened by further analysis of conformational sensitivity, transferability across chemical domains, and validation beyond this single property.

---

### Decision · Program_Chairs · 2026-03-02

Accept (Poster)